# RMR-Related *MAP2K6* Gene Variation on the Risk of Overweight/Obesity in Children: A 3-Year Panel Study

**DOI:** 10.3390/jpm11020091

**Published:** 2021-02-02

**Authors:** Myoungsook Lee, Yunkyoung Lee, Inhae Kang, Jieun Shin, Sungbin R. Sorn

**Affiliations:** 1Department of Food & Nutrition, and Research Institute of Obesity Sciences, Sungshin Women’s University, Seoul 01133, Korea; 2Department of Food & Nutrition, and Interdisciplinary Graduate Program in Advanced Convergence Technology and Science, Jeju National University, Jeju 63243, Korea; lyk1230@jejunu.ac.kr (Y.L.); inhaek@jejunu.ac.kr (I.K.); 3Department of Biomedical Informatics, College of Medicine, Konyang University, Daejeon 353565, Korea; 598977@hanmail.net; 4Department of Bioinformatics & Statistics, Graduate School, Korea National Open University, Seoul 03087, Korea; sorn7@kaist.ac.kr

**Keywords:** child obesity, Resting Metabolic Rate (RMR), MAP2K6 (MEK6), energy expenditure, waist circumferences (WC), Systolic Blood Pressure (SBP), dietary fat intake

## Abstract

From a pilot GWAS, seven *MAP2K6 (MEK6)* SNPs were significantly associated with resting metabolic rate (RMR) in obese children aged 8–9 years. The aim of this study was to investigate how RMR-linked *MEK6* variation affected obesity in Korean children. With the follow-up students (77.9%) in the 3-year panel study, the changes of the variables associated with obesity (such as anthropometrics, blood biochemistry, and dietary intake) were collected. After the *MEK6* SNPs were screened by Affymetrix Genome-Wide Human SNP array 6.0, the genotyping of the seven *MEK6* SNPs was performed via SNaPshot assay. As the prevalence of obesity (≥85th percentile) increased from 19.4% to 25.5%, the rates of change of the variables RMR, body mass index (BMI), waist circumference (WC), systolic blood pressure (SBP), and dietary intake (energy and carbohydrate intakes) increased. The rate of overweight/obesity was higher in all mutant alleles of the seven *MEK6* SNPs than it was in the matched children without mutant alleles. However, over the 3-year study period, RMRs were only significantly increased by the mutants of two single nucleotide polymorphisms (SNPs), rs996229 and rs756942, mainly related to male overweight/obesity as both WC and SBP levels increased. In the mutants of two of the SNPs, the odds ratio of overweight/obesity risk was six times higher in the highest tercile of fat intake and SBP than those of the lowest tercile. For personalized medicine to prevent pediatric obesity, SBP, WC, and dietary fat intake should be observed, particularly if boys have mutants of *MEK6* SNPs, rs9916229, or rs756942.

## 1. Introduction

The number of children and adolescents with obesity has risen more than 10-fold (from 11 to 124 million) in the past 4 decades. The increasing prevalence of childhood obesity is a worldwide trend, which results in early-onset adult obesity as a consequence of metabolic imbalance [1]. Most of the previous approaches used to elucidate the mechanisms of obesity were convergent with the four regulation pathways: food intake, fat absorption, adipogenesis or lipolysis, and thermogenesis [2,3]. Therefore, the development of medicines or functional products focused on these four pathways. However, since those efforts concluded in unsatisfactory outcomes, the need for a new approach, such as the regulation of energy balance, is crucial. The research showing that energy restriction aids in weight loss is well established, but the research on energy expenditure for weight loss is limited to studies on exercise. The resting metabolic rate (RMR), including basal metabolic rate, explains the most substantial proportion (60–70%) of total daily energy needs in individuals [4]. Consequently, the environmental factors related to the RMR that control energy balance have been investigated. Genetics, age, muscle mass, gender, physical activities, hormonal factors, drugs, and diet may influence the balance of energy metabolism with RMR [5,6]. However, the contribution of a low RMR to the etiology of obesity is still controversial [7]. Based on several studies (including the Pima Indian case) that have shown that RMR has a strong genetic component, RMR might be a predictor of subsequent weight change in subjects with mutant genes that control RMR by racial or gender difference [8]. In particular, BMI-associated RMR study is essential to predict early-onset adult obesity in the development period of children where the most growth occurs.

From our pilot genome-wide association study (GWAS), we found that mitogen-activated protein kinase 6 (*MAP2K6* or *MEK6*), a significant gene on chromosome 17, was commonly associated with both BMI and RMR in Korean boys and girls. The roles of MAPKs-p38 kinase (p38), Jun amino-terminal kinases (JNK), and extracellular signal-regulated kinase (ERK) on the cellular function and gene expression response to diverse stimuli (such as stress, heat shock, and cytokines) have become widely known [9,10,11]. As a potent upstream activator of p38 both in vitro and in vivo, *MEK6* induces the initial differentiation of 3T3-L1. *MEK6* promotes adipogenesis by the regulation of peroxisome proliferator-activated receptor gamma (PPAR-γ) and CCAAT/enhancer-binding protein beta (C/EBP-β) [10]. In human genetic studies, 28 genes encoded protein to be differently expressed in the livers of obese women, and *MAP2K6* was screened as a gene that contributed to signal transduction [12]. Sequence variations of *MAP2K6* with ASK1 lead to gender-specific changes in the levels of p38-regulated proteins, contributing to delaying effects in the age of onset of Huntington’s disease [13]. However, it is unknown whether *MEK6* is involved in the prevalence of obesity as well as its mechanisms.

Based on the pilot GWAS, the purposes of this study were to identify how RMR affects obesity risk variables according to the different *MEK6* SNPs and to provide the basic mechanism to control energy balance in future study.

## 2. Materials and Methods

### 2.1. Study Design

Three hundred and seventeen children (aged 8–9 years, 3rd grade students from eight elementary schools (Guro-gu, Seoul, Korea)) were recruited during the 2009 annual health check period. With a 77.9% follow-up (F/U) rate in the 3-year panel study, 247 of the 317 children, the same students in 2012 now in 6th grade, were included as final subjects. As a baseline, children who had medical problems noted by the doctors or children whose parents did not agree with F/U study from 2009 to 2012 were excluded; otherwise, all children with a consent form were included. During the F/U period, we lost 70 subjects for personal reasons; fear of the blood tests, sudden illnesses, non-participation in regular medical tests, and parental refusal without reason. At the baseline, the randomly selected 107 children (58 boys and 59 girls) were involved in the GWAS study to find RMR genes related to BMI ≥ 85th percentile. Seven *MEK6* SNPs were found in this period that were related to the *MEK6* gene, and we investigated how RMR affected the change rates ((2012–2009)/2012) of the anthropometrics, blood chemistry, and dietary intake variables. The flowchart of this 3-year panel study is shown in Figure 1. The IRB of Human Research of Korea University, Guro Hospital (No: GR0837-001), approved this study.

### 2.2. DNA Extraction, GWAS and MEK6 SNPs Genotyping

Blood genomic DNA was extracted using a DNA extraction kit (LaboPass Blood MiniKit, Cosmo Genetech, Korea). DNA quantity and quality were evaluated with a spectrophotometer (Smartspec plus, Bio-Rad Laboratories, Richmond, CA, USA), gel analysis (2% agarose gel, Duchefa Biochemie, Haarlem, Netherlands) (which was conducted with Redsafe Nucleic Acid Staining Solution (iNtRON Biotechnology Inc., Seoul, Korea)), and PCR amplification (iCycleriQ PCR system, BIO-RAD). All DNA stocks were diluted to 20 ng μL^−1^ with a Tris–EDTA buffer (pH 8.0) and stored at −20 °C. For GWAS analysis, the criteria of the case and control for both RMR (Low; <1055.8 kcal/day as a median) and BMI (obesity + overweight, ≥85th percentile; >19 for boys, >18.56 for girls) were determined in 107 pilot subjects. Using an Affymetrix genome-wide human SNP array 6.0 in DNALink (DNALINK, Inc, Seoul, Korea), *MEK6* was found to be a significant gene (*p* < 10^−4^), and seven *MEK6* SNPs were found in the period related to the *MEK6* gene (64946482-65168553) using the linkage disequilibrium plot. The genotyping of all seven *MEK6* SNPs (rs12603937 (G/T as wild/mutant), rs11654541 (G/T), rs1051273 (G/A), rs2285601 (T/A), rs2285600 (G/A), rs9916229 (C/A), and rs756942 (C/T)) was performed via SnaPshot assay according to the manufacturer’s instructions (ABI PRISM SNaPShot Multiplex kit, Foster City, CA, USA). The genomic DNA flanking the SNPs was amplified using PCR reaction with forward and reverse primer pairs. (Appendix A) Analysis was carried out using Genemapper SW (version 4.0; Applied Biosystems (Foster City, CA, USA)).

### 2.3. Variables of Anthropometric, Biochemistry, and Dietary Intake

Standing height (Ht), weight (Wt), and waist circumference (WC) were measured with the standard techniques. Systolic and diastolic blood pressures (SBP & DBP) were measured with an automatic BP calculator (NISSEI, Hanishina-gun, Nagano-ken, Japan). BMI was calculated by dividing the subject’s weight (Kg) by the square of their height (m^2^). Using the Application of 2010 Korean National Growth Charts, boys and girls were divided into two groups; the overweight/obesity (OB, BMI ≥ 85th percentile) group and the non-obesity (non-OB, BMI < 85th percentile) group. The predictive RMR equation of Harris & Benedict (HR-RMR; 655.1 + 9.6Wt + 1.9Ht−4.7A; girls, & 66.5 + 13.8Wt + 5Ht−6.8A; boys, as A; age) was used for this cohort study. Whole blood used for genotyping in EDTA-treated tubes and serum used for other blood biochemistry were collected after 12 hr fasts. Lipid profiles, such as total cholesterol (TC), TG, and HDL-cholesterol (HDLc) levels and fasting serum glucose (FBS) levels, were measured using an autoanalyzer (Ekachem DTSC module, Johnson & Johnson, New Brunswick, NJ, USA). LDL-cholesterol (LDLc) levels were calculated as described by the Friedewald and Lauer equation. Serum insulin levels were measured with an ELISA kit (Media, Co. Seoul, Korea), and the Matthew equation calculated the homeostasis model assessment of insulin resistance (HOMA-IR) values, fasting glucose (mmol/L)/22.5 × insulin (µL/mL). Aspartate transaminase (AST) and alanine transaminase (ALT) were measured with an ELISA kit (Human diagnostic worldwide, Wiesbaden, Germany). The dietary intakes for three days, two weekdays, and one weekend day were recorded according to the 24 h dietary recall method by the trained interviewers. Parents, guardians, or dieticians in the schools cross-checked the food records. Food records were finally included in the dietary analysis after excluding incomplete, undetectable, or unreliable records. CAN-Pro 4.0 (Korean Nutritional Society, Seoul, Korea) was used to analyze nutrients based on the food records quantitatively. Environmental data such as dietary habits (starvation, fast-food intake, eating out times, etc.) and the degree of physical activity were produced through the use of questionnaires.

### 2.4. Statistics

For the analysis of GWAS, the Hardy–Weinberg equilibrium (HWE) was assessed using the χ^2^ test. All datasets were filtered to exclude samples or SNPs with > 5% missing values, variants with < 5% minor allele frequencies (MAFs), and samples deviating from the HWE using PLINK, a whole-genome data analysis toolset. Quantitative trait association analyses for RMR and BMI in the 107 children (RMR < 1055.8 kcal/day and BMI > 85th percentile) were performed. BMI and RMR were analyzed as continuous traits (separately in obese and non-obese subjects) with linear regression in PLINK, using dominant, codominant, and recessive models. Genotype and allele frequencies were compared between groups using the chi-square test, the Cochran Armitage trend test, or the Jonckheere–Terpstra test as appropriate. Data analysis was performed using SAS 9.1.3 (SAS Inc., Cary, NC, USA) and PLINK [14]. Continuous variables were expressed as mean ± SD, and differences between OB and non-OB groups were assessed using a Student’s t-test. Categorical variables were represented as percentages and tested using the χ^2^ test. Using SAS 9.3 (SAS 9.3, SAS Institute, Cary, NC, USA), the rates of change in all of the variables were analyzed in terms of 2009 data vs. 2012 data by a paired t-test (PROC TTEST). To find RMR differences between the dominant or recessive alleles of *MEK6* SNPs, we tested the Levene’s normal distribution. If the RMRs were not normally distributed in genotypes and alleles of SNPs, Welch’s t-test was used. An independent t-test examined the differences in variables according to the alleles of seven *MEK6* SNPs, and RMR differences according to the genotypes of seven SNPs were analyzed via one way-ANOVA test. Linear regression analysis was performed for BMI as a response and RMR. The stepwise multiple regression analysis screened for the risk factors that increased BMI according to the seven *MEK6* SNPs. After the residual energy intakes adjusted all nutrient data collected from subjects with the simple linear regression, the data were analyzed by gender and the degree of overweight/obesity. Odds ratio (OR) statistics were defined to determine the strength of the association between the risk of overweight/obesity and the changes of variables according to the seven *MEK6* SNPs.

## 3. Results

### 3.1. Changes of Variables over 3 Years

The rate of overweight/obesity in 2012 (25.5%) was increased by 6.1% compared to 2009 (19.4%). RMR was significantly increased as a result of increasing BMI over three years, with a positive correlation between BMI and RMR (BMI = 15.2 + 0.0042 RMR, R^2^ = 30.0%, *p* = 0.0013) (Figure 2). Although both the BMI and RMR indicators were calculated by Wt and Ht, the correlations between RMR and Wt or Ht were 0.7042 or 0.7730, respectively, but the correlations between BMI and Wt or Ht were not significant. We carefully proposed that adjusting RMR with Ht or Wt is unnecessary for children’s obesity studies. At the baseline, the overweight/obesity (OB) group showed significantly higher levels of RMR, BMI, WC, BP, TG, dietary sodium, and riboflavin intake compared to the non-OB group. In 2012, RMR, BMI, WC, BP, and energy intake were higher in the OB group than in the non-OB group. Plasma HDLc and niacin intake were lower in both 2009 and 2012 in the OB group. With the reduction of the total energy intake in the OB group from 2012, other dietary intakes exhibited similar patterns (Table 1). For 3 years, the rates of change of RMR, BMI, WC, and HDLc increased, while those of LDLc and fasting blood glucose (FBS) were reduced. The rates of change of dietary intakes such as carbohydrates (CHO), monounsaturated fatty acids (MUFA), riboflavin, and Vitamin B_6_ were increased. Otherwise, the rates of change of total fatty acids (TFA) (including SFA and polyunsaturated fatty acids (PUFA)), fiber, minerals (Ca, Zn, folate, K), and vitamins (A, B_1_, C, and niacin) significantly decreased (Table 1).

### 3.2. Association with Overweight/Obesity and The Types of MEK6 SNPs

The minor allele frequencies (MAF) of seven *MEK6* SNPs, shown in allele number (wild/mutant), of rs12603937 (G/T), rs11654541 (G/T), rs1051273 (G/A), rs2285601 (T/A), rs2285600 (G/A), rs9916229 (C/A), and rs756942 (C/T), in our subjects were 25.4%, 26.2%, 26.1%, 25.8%, 25.9%, 35.5%, and 30.5%, respectively. According to the allele’s frequencies in 1000G DB, each MAF of the seven *MEK6* SNPs were 28.5%, 31.9%, 31.9%, 31.9%, 31.8%, 38.8%, and 37.6%, respectively, with similarly high MAFs in rs9916229, and rs756942 in this population. The range of genotype frequencies for which the Hardy–Weinberg equilibrium is satisfied is shown as the curve within the diagram (Appendix A). Therefore, statistical analysis for the minor alleles’ effects on overweight/obesity prevalence and obesogenic environments could be valid (Figure 3A). Since the MAFs of all SNPs were within the range of 25 to 35%, the relative frequencies (RF) of OB (overweight/obesity) in the mutant allele of all seven *MEK6* SNPs were significantly higher than those in the wild allele (Figure 3B) (Appendix A). In boys, the higher RFs of OB in the mutant allele were shown in two SNPs, rs9916229 (27.9%) and rs756942 (28.4%). However, the other five SNPs had significant differences in RF of OB (>56.3%) in girls, except for rs9916229 (46.3%) and rs756942 (48.3%). The mutants of two SNPs, rs9916229, and rs756942, have been shown to demonstrate gender differences in OB prevalence.

### 3.3. RMRs According to the Types of MEK6 SNPs

We confirmed that changes in RMR over 3 years were positively increased by up to 20 Kcal in all wild alleles of the seven SNPs. The means of RMRs (1152.42 Kcal/d) in all mutant alleles of the seven SNPs were higher than in their corresponding wild types (1125.8 Kcal/d). Interestingly, the RMRs of only two of the *MEK6* SNPs (rs9916229 and rs756942) were significantly different among wild homozygotes, heterozygotes and mutant homozygotes at the baseline (Figure 4A). In the other five *MEK6* SNPs, the RMR differences between wild and mutant alleles were significant, but not in genotypes. Combined wild or mutant homozygotes of two SNPs (rs9916229 and rs756942) and RMRs in mutant homozygotes were significantly increased for 3 years compared to those in the wild homozygotes (Figure 4B). Although the differences in BMI over 3 years were not significantly changed in both SNPs, the BMI in children with mutant homozygotes falls under the criteria of overweight/obesity (≥85 percentiles) for both years. BMIs in subjects with two mutant homozygotes (*n* = 14) of rs9916229 and rs756942 were significantly higher than those in subjects with two wild homozygotes (*n* = 110). (Figure 4C) It was interesting that the number of mutant homozygotes in rs9916229 and rs756942 were 21 and 16, respectively, but the number of mutant homozygotes of both rs9916229 and rs756942 was 14. If the children had one of the mutants of the two SNPs, the probability of overweight/obesity was 66–88%. We found two SNPs (rs9916229 and rs756942) were significantly related to gender difference in their association between RMR and the prevalence of pediatric obesity.

### 3.4. The Changes in Risk Factors on the Risk of Overweight/Obesity in Alleles of MEK6 SNPs

Using stepwise multiple regression analysis, we found that the risk variables in mutant alleles indicating a high prevalence of overweight/obesity were not significant in the wild allele in both SNPs (rs9916229 and rs756942). Variables such as WC, SBP, fat, and fiber intake, whose levels increased for 3 years, were significant risk factors in increasing BMI (*p* < 0.001) in the minor allele of rs9916229. In the case of the minor allele of rs756942, variables of WC, SBP, fat, and folate intake were risk factors. (Table 2). WC was a unique factor, increasing the risk of overweight/obesity for children with both major and minor alleles of the two SNPs. If children aged 8–9 years old have the minor alleles of either rs9916229 or rs756942, WC, SBP, and fat intake are the most influential risk factors in terms of BMI increase over 3 years despite their differential dietary effect on the two SNPs. After the rates of change of all variables were categorized by their tercile values (Q1; low, Q2; medium, Q3; high), the relative risks of overweight/obesity (odds ratio; ORs) were compared between Q2 or Q3 and Q1 according to alleles of all *MEK6* SNPs. In all subjects, the risk of overweight/obesity was reduced by increasing HDLc and decreasing LDLc, energy intake, and fat intake (Data not shown). However, the OR for the risk of overweight/obesity was almost six times higher in the highest rate of change of fat intake (Q3 > 1.11) than in the lowest one (Q1 < 0.8) in the minor alleles of two *MEK6* SNPs (rs9916229 (6.37, CI;1.47~27.58) and rs756942 (6.25, CI;1.44~27.05)). The OR in the Q2 of fat intake was almost four times higher than that of Q1 in both rs9916229 (4.29, CI;1.09~16.79) and rs756942 (4.11, CI;1.05~16.05). The change rates of SBP in Q3 (>1.05) also increased the risk of overweight/obesity six times higher than in Q1 (<0.92) in both rs9916229 (6.98, CI;2.95~16.47) and rs756942 (6.75, CI;2.756~16.55). The significant finding was that two *MEK6* SNPs (rs9916229 and rs756942) were strongly associated with the prevalence of overweight/obesity in boys only (Figure 5). In conclusion, if boys have a mutant allele of *MEK6* SNPs such as rs9916229 or rs756942, reducing dietary fat intake is recommended to prevent early-onset obesity and adult obesity. Periodic control of SBP is also necessary.

## 4. Discussion

In the pilot GWAS, we found that the *MAP2K6* (*MEK6,* Ch#17) gene was related to RMR in children with obesity. Using the mutant alleles of seven *MEK6* SNPs, we found that they were strongly associated with high overweight/obesity prevalence, and the mutants of two SNPs (rs996229 and rs756942) were particularly related to boys’ obesity, including overweight. Since the risk of overweight/obesity was six times higher in the highest tercile of fat intake and SBP than in the lowest tercile in minor alleles of rs996229 and rs756942, *MEK6* provides new insight into the role of the metabolic rate in controlling the energy balance.

The prevalence of obesity increased twice (8.7% to 15.0%) during 10 years (2007 to 2017) in children aged 6–18 years [15]. The prevalence of overweight/obese children aged 2–19 increased from 1998 to 2012, from 15.2% to 25.5% [16]. A more critical finding was that 60% of Korean children with obesity aged 10–18 years had at least one CVD risk factor [17]. Therefore, the discovery of the environmental risk factors of childhood obesity and the methodology of diagnosis or treatment is essential because many kinds of research have shown that childhood obesity associated with early-onset adult chronic diseases has increased morbidity and mortality [18]. At the baseline, we found that obese children seemed to have inherited obesity judging by the BMI of their parents, but it was not correlated with parents’ education and income, health supplement intake, time of TV watching, and snack intake [19]. Therefore, the environmental factors, including nutrigenomics related to RMR in the control of the energy balance, should be considered, rather than dietary habits, physical activities, or hormones [4].

RMRs increased by 20% for 3 years, corresponding to an increase in the BMI in the growth curve. The 2015 Korean Dietary Reference Intake (KDRI) for the total energy requirement (TEE), measured by double-labeled water, suggested that TEEs were 2100 Kcal/day for 9–11 year olds and 2400 Kcal/day for 12–14 year olds. The values of HR-RMRs in children 8–11 years old in this study were almost 60% of TEE a day. In Kim’s study, TEEs in children aged 9–11 years were 1925–1930 Kcal/day and RMRs in children were 1220–1240 Kcal/day, which were 63% of TEE [20]. Kim et al. also reported that HR–RMR was close to the KDI formula following the IOM equation for Korean elementary students [21]. HR–RMRs (1240.9 ± 147.4 Kcal/day) in 102 local children (9–13 years) were very similar to those shown by our results (8–11 years, 1138.8 ± 123.4 Kcal/day).

RMRs were increased in the OB group compared to the non-OB groups. Based on the meta-analysis, whether the cause of the low RMR for which given body size and composition is genetic or acquired, the existence of a low RMR is likely to contribute to a high rate of weight regain [7]. Formerly, obesity had a 3–5% lower mean relative RMR than that of control subjects, and the difference could be explained by a low RMR being more frequent among the formerly obese than among the non-obese. The insulin and HOMA-IR levels were more favorable in people with a high RMR, but the sleeping RMR was higher in 560 Pima Indians with diabetes and impaired glucose tolerance [22,23]. Compared RMR in obese adults (fat% of 33.6) and non-obese adults (fat% of 20.4) aged 20–30 years, the RMR (1550 kcal/d) was higher in obese individuals than non-obese individuals (1421 kcal/d) [24]. In China, a new predictive RMR (=17.4 × LogFFM + 11.4 × Conicity index-2.4 × Centrality index) was established for obese children based on the high correlation between measured RMR and FFM [25]. Absolute RMR (obtained using a ventilated hood system) was not different in Asian and White adults, but low RMR in Asians can be attributed to body composition as fat-free mass (FFM) [26]. To predict the best RMR prediction for obese or non-obese children, we considered RMR body composition models not to be constant with gender, age, and ethnic differences [8,27,28].

Using the respiratory quotient (RQ) as an indirect method of RMR, dietary sources of energy expenditure can be traced, but RMR and RQ did not show consistency in weight gain in the previous study [29]. In the study of 169 Quebec families, GWAS for quantitative trait loci (QTL) contributing to the variability in RMR and RQ was reported for obesity, type 2 diabetes, and metabolic syndrome [30,31]. They found the linkage to RMR on chromosomes 3q26.1 (lod = 2.74), 1q21.2 (2.44), 22q12.3 (1.33), and QTL-influencing RQ was found on chromosomes 12q13 (1.65) and 14q22 (1.83). It was interesting that RMR linkage on 1q21.2 showed links between sleeping metabolic rate or energy metabolism in type II diabetes for Pima Indians [32]. The 1q21 region includes the interleukin six receptor, complexes I and II in the electron transport system on mitochondria, and so on. The Hellwege adult study investigated single variant analysis for RMR identified significant loci on chromosomes 15, 1, 17, and 5 in African- and European-American children and adults [33]. They found the most significant locus was SH3D21 (*p*-value 2.01 × 10^−4^), and they found nominal evidence for association of BMI-associated loci with RMR with rs35433754 (TNKS) within the reported genes for all obesity-related loci from the GWAS. Compared to the Quebec and Hellwege adult studies, the genes found through GWAS related to RMR and BMI in children were different because of the different life cycle stages. We had the GWAS study for Korean women related to the basal metabolic rate and BMI, but there were significant differences between the Hellwege and Quebec studies because of gender or racial differences [34]. In our in vitro study, the genes related to fat accumulation (PPARr, C/EBPa, and aP2) and inflammatory cytokines (IL-1β, TNF-α, leptin/adiponectin ratio, etc.) were highly expressed in *MEK6*-overexpressed 3T3-L1 cells. The time taken to reach the maximal oxygen consumption in *MEK6*-overexpressed 3T3-L1 cells was slower than that in control (non-published paper). It was a clue that RMR may affect energy consumption with the variation of the MEK6 gene.

This is the first report stating that dietary intakes such as fat, fiber, Ca, and folate are involved in BMI variation with the mutant alleles of RMR-linked *MEK6* SNPs, rs9916229 and rs756942. Folate activates MAPK ERK2 (MEK6), which requires activation of the G protein and adenylyl cyclase to stimulate the intracellular regulatory pathways [35]. There was no reduction in dietary fat, but the carbohydrate-to-protein ratio is a suitable energy expenditure model to increase RMR with the sustaining FFM [36]. Diets with a low carbohydrate (50%) or low glycemic index (GI) did not change RMR and weight loss (FM) compared to diets with high carbohydrate (70%) and high GI such as dietary fiber in 17 weeks RCT and weight regained 12 months later [37]. Some studies have suggested that dietary variation in protein, carbohydrate, and low GI have attenuated RMR and body composition, but dietary effects on the metabolic rates are still controversial [38,39]. There were many studies stating that dietary intake affects the control of energy intake, but the research on whether dietary intake is related to RMR in the variation of the energy expenditure should be considered in the future.

In conclusion, for boys with any mutant allele of *MEK6* SNPs, such as rs9916229 and rs756942, we suggest the need for a therapeutic guideline, as the aspects of personalized medicine such as dietary fat intake and SBP should be controlled to prevent early-onset of chronic disease for children.

As for limitations of this study, even though we recognized the importance of body composition (i.e., fat (%) or lean body mass (%)), the measurement of body composition in children 8~12 years old requires DEXA, which was not available for economic, physical, and ethical reasons. Secondly, we used the predicting RMR method instead of the indirect calorimetry method because of several reasons, such as parents’ disagreement with long-time consumption testing in children and funding. Thirdly, the adjusted methods (repeated measures correlation; rmcorr) for within-individual association to avoid type-I error were not performed. However, the strength of our study is that it is the first report to find how the variation of RMR-related gene *MEK6* modulates overweight/obesity in Korean children. Therefore, we may suggest that *MEK6*, as an RMR related gene, provides new insights into energy balance, and it will serve as a reference for validation and generalization in future studies of children.

## Figures and Tables

**Figure 1 jpm-11-00091-f001:**
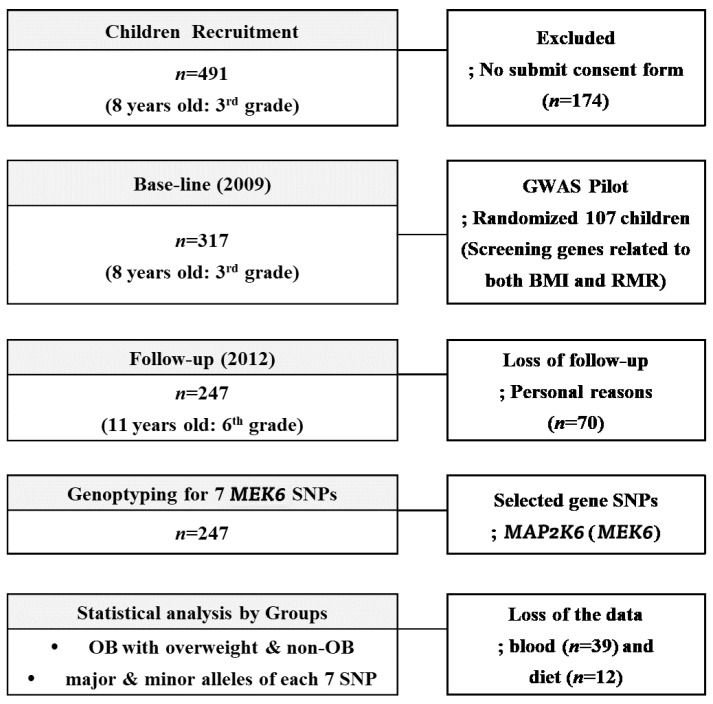
The flowchart of the 3-year panel study accounting the subjects.

**Figure 2 jpm-11-00091-f002:**
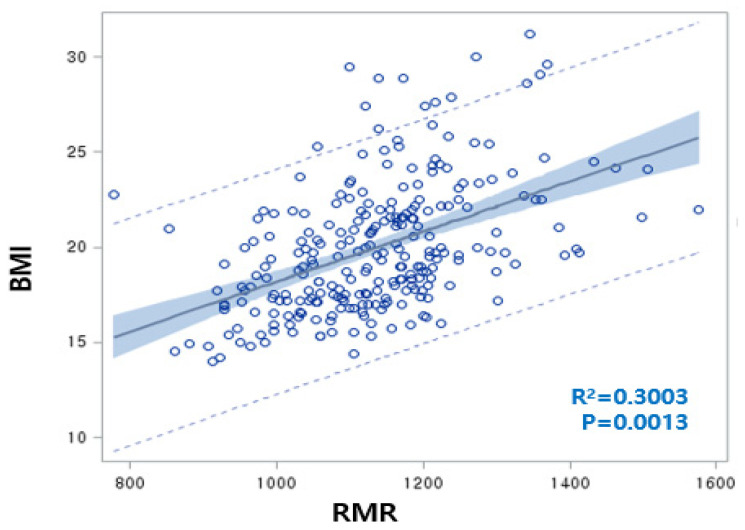
Linear regression analysis conducted regarding body mass index (BMI) (as an obesity marker) and resting metabolic rate (RMR) (BMI = 15.2 + 0.0042 RMR, R^2^ = 30.0%, *p* = 0.0013). The correlation between RMR and weight (Wt) or RMR and height (Ht) was 0.7042 or 0.7730, respectively.

**Figure 3 jpm-11-00091-f003:**
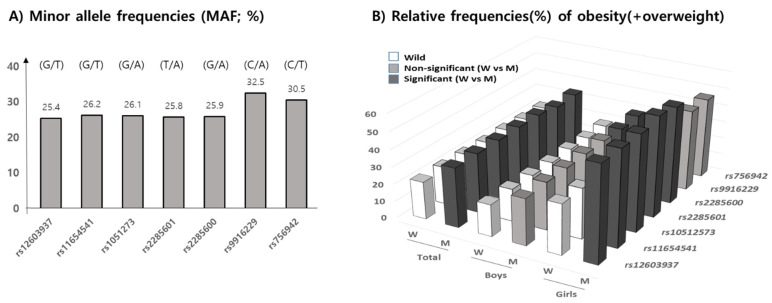
Minor allele frequencies (MAF, %) according to 7 MEK6 SNPs are shown, with descriptions of the alleles (wild/mutant). (**A**) The relative frequencies (RF) of obesity, including overweight (%), (**B**) in both the wild and mutant allele of seven SNPs in total, for both boys and girls, are shown. Black bars represent significant differences in RF between wild and mutant alleles (*p* < 0.05), but the gray bars were not significant. Only two SNPs, rs9916229 and rs756942, show the gender differences in terms of RF of overweight/obesity.

**Figure 4 jpm-11-00091-f004:**
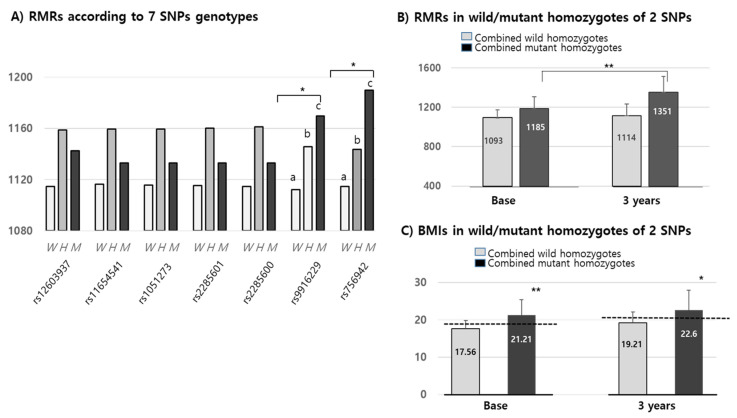
Mean values (Kcal) of RMR & BMIs in MEK6 SNPs genotypes. (**A**) RMRs according to the seven 7 SNPs genotypes of the MEK6 gene (such as wild (W), hetero (H) & mutant (M)) of each SNP. SD bars are not described in this figure because the mean of the SDs (≒129.2 Kcal) was higher than the *Y*-axis range. (**B**) RMRs in mutant homozygotes of two SNPs (rs9916229 and rs756942) were significantly increased at both the baseline and 3 years later compared to those in wild homozygotes. (**C**) BMIs in mutant homozygotes of two SNPs (rs9916229 and rs756942) were significantly increased at both the baseline and 3 years later compared to those of wild homozygotes. The mean values of BMI ≥ 85th percentile at both 2009 and 2012 are shown in the dotted line (baseline: 18.56–19, 3 years; 20.59–21). Significant differences (*; *p* < 0.05, **; *p* < 0.001) are described with an alphabetic subscript.

**Figure 5 jpm-11-00091-f005:**
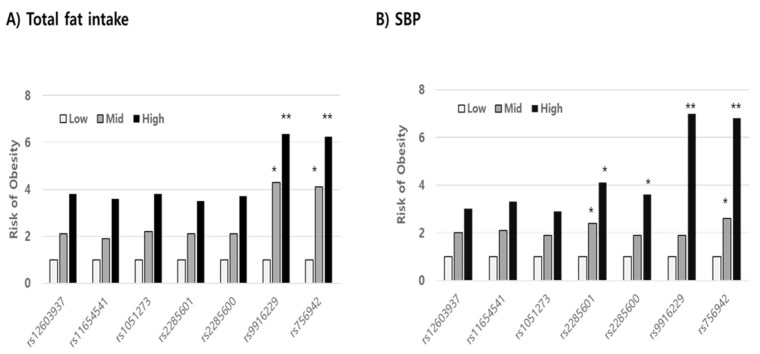
Comparing the relative risk of overweight/obesity in the tercile values (low, medium, high; Q1, Q2, Q3) of variables related to obesity. The odds ratios (ORs) in both the Q3 levels of dietary fat intake (**A**) and SBP (**B**), significantly, were six times higher than those in the Q1 levels in minor alleles of specific *MAP2K6* SNPs such as rs9916229 and rs756942 (Significance compared to Q1; *; *p* < 0.05, **; *p* < 0.01).

**Table 1 jpm-11-00091-t001:** Difference of variables between two groups (non-OB and obese) and rates of change of variables between 2009 and 2012 in the same follow-up (F/U) students.

Variables	2009 ^a)^					2012					Change Rates (2012–2009/2012)		
		Non-OB		OB (Overweight/Obesity)			Non-OB		OB (Overweight/Obesity)				
	N	Mean ± Std		Mean ± Std	*p*-Value ^b)^	N	Mean ± Std		Mean ± Std	*p*- Value ^b)^	N	Mean ± SD	*p*- Value ^c)^
RMR (Kcal/d)	199	1088.7 ± 98.3	48	1133.9 ± 108.4	*	184	1261.3 ± 128.6 ^¶^	63	1481.6 ± 143.2 ^¶^	**	247	0.19 ± 0.04	***
BMI (kg/m2)	199	17.1 ± 1.7	48	22.8 ± 1.9	***	184	18.4 ± 2.0 ^¶^	63	24.5 ± 2.3 ^¶^	***	247	0.10 ± 0.10	***
Waist (cm)	199	56.6 ± 5.4	48	70.8 ± 6.1	***	178	65.0 ± 6.3 ^¶^	62	79.4 ± 7.5 ^¶^	***	240	0.16 ± 0.15	***
SBP (mmHg)	198	109.8 ± 17.0	48	116.9 ± 20.4	*	184	109.4 ± 9.3	63	117.8 ± 11.1	***	246	0.11 ± 1.06	-
DBP (mmHg)	198	71.5 ± 10.7	48	76.8 ± 11.6	**	184	68.5 ± 7.6 ^¶^	63	73.8 ± 8.6	***	246	−0.02 ± 0.18	***
TC (mg/dL)	199	165.7 ± 27.7	48	169.6 ± 36.0	-	156	166.5 ± 24.3	52	168.5 ± 25.8	-	208	0.02 ± 0.19	-
TG (mg/dL)	199	79.3 ± 43.3	48	115.5 ± 52.9	***	156	88.6 ± 47.8	52	100.9 ± 45.6	-	208	0.24 ± 0.66	-
LDL (mg/dL)	199	98.0 ± 27.4	48	102.5 ± 35.5	-	156	86.3 ± 19.7 ^¶^	52	94.2 ± 24.0	*	208	−0.07 ± 1.10	***
HDL (mg/dL)	199	51.9 ± 10.9	48	43.9 ± 10.5	***	156	62.4 ± 13.5 ^¶^	52	54.1 ± 12.5 ^¶^	***	208	0.21 ± 0.20	***
FBS (mg/dL)	194	79.9 ± 8.3	48	80.7 ± 6.7	-	156	75.9 ± 8.2*	52	75.1 ± 8.7 *	-	208	−0.05 ± 0.14	***
Energy (kcal)	197	1694.2 ± 335.8	47	1628.3 ± 332.9	-	179	1997.4 ± 558.6 ^¶^	59	1703.1 ± 481.7	***	235	0.18 ± 0.39	***
CHO (g)	197	241.7 ± 21.0	47	242.1 ± 18.2	-	179	246.9 ± 43.2	59	249.9 ± 35.1	-	235	0.04 ± 0.20	*
Protein (g)	197	65.7 ± 12.4	47	63.2 ± 7.2	-	179	66.9 ± 48.9	59	66.4 ± 22.2	-	235	−0.07 ± 0.27	***
Fat (g)	197	52.3 ± 7.7	47	51.8 ± 7.6	-	179	51.1 ± 15.5	59	52.5 ± 13.5	-	235	0.13 ± 1.63	-
Cholesterol (mg)	197	314.9 ± 111.6	47	339.9 ± 106.6	-	175	275.3 ± 167.9 ^¶^	58	302.5 ± 177.2	-	235	0.01 ± 0.33	-
TFA (g)	197	30.6 ± 8.2	47	32.1 ± 9.2	-	175	26.03 ± 14.4 ^¶^	59	28.1 ± 14.1	-	230	−0.01 ± 0.72	**
SFA (g)	197	12.0 ± 4.0	47	12.5 ± 4.7	-	172	10.7 ± 6.2 ^¶^	58	11.6 ± 5.6	-	231	−0.07 ± 0.59	***
MUFA (g)	197	10.8 ± 3.2	47	11.3 ± 3.6	-	174	9.5 ± 5.8 ^¶^	59	10.3 ± 5.8	-	227	0.01 ± 0.76	**
PUFA (g)	197	7.38 ± 2.3	47	7.6 ± 2.3	-	177	6.1 ± 3.4 ^¶^	59	6.3 ± 3.2 ^¶^	-	230	−0.02 ± 0.69	**
Fiber (g)	197	15.0 ± 2.6	47	15.7 ± 3.3	-	179	13.8 ± 4.0 ^¶^	59	14.2 ± 2.9 ^¶^	-	233	−0.07 ± 0.63	***
Calcium (mg)	197	597.3 ± 151.9	47	600.5 ± 107.1	-	179	498.4 ± 206.5 ^¶^	59	547.02 ± 234.71	-	235	−0.12 ± 0.39	***
Fe (mg)	197	11.3 ± 2.6	47	12.1 ± 3.6	-	179	11.5 ± 5.4	59	11.8 ± 9.4	-	235	−0.03 ± 0.26	*
Phosphate(mg)	197	965.4 ± 144.1	47	947.7 ± 130.1	-	179	923.0 ± 230.5 ^¶^	59	940.6 ± 241	-	235	0.07 ± 0.73	-
Zinc (mg)	197	8.1 ± 1.0	47	8.0 ± 0.91	-	179	7.9 ± 1.7	59	7.8 ± 1.6	-	234	−0.07 ± 0.33	***
Folate (µg)	197	208.3 ± 51.5	47	219.8 ± 56.4	-	178	208.0 ± 96.2	59	193.1 ± 61.4 ^¶^	-	235	−0.06 ± 0.33	***
Na (mg)	197	3449.4 ± 792.2	47	3801.9 ± 976.2	**	178	3147.2 ± 996.4 ^¶^	59	3229.3 ± 992.8 ^¶^	-	235	−0.02 ± 0.25	-
K (mg)	197	2356.4 ± 442.2	47	2331.0 ± 470.8	-	179	2144.5 ± 682.1 ^¶^	59	2152.9 ± 638.4 ^¶^	-	235	−0.02 ± 0.62	*
Retinol (µg)	197	178.8 ± 79.1	47	178.1 ± 65.6	-	175	168.7 ± 120.0	59	166.0 ± 120.6	-	231	0.08 ± 0.85	-
β-carotene (µg)	197	2804.6 ± 1064.6	47	2888.6 ± 1054.7	-	178	2591.9 ± 2182.3	59	2477.3 ± 1404.4	-	234	0.03 ± 1.02	-
Vit A (µgRE)	197	698.2 ± 231.7	47	747.8 ± 9268.0	-	179	645.98 ± 397.78	59	637.1 ± 271.4	-	235	−0.01 ± 0.36	*
Vit B1 (mg)	197	1.2 ± 0.4	47	1.3 ± 0.3	-	179	1.2 ± 0.4	59	1.2 ± 0.3 ^¶^	-	235	−0.10 ± 0.32	***
Vit B2 (mg)	197	1.3 ± 0.3	47	1.4 ± 0.4	*	179	1.2 ± 0.4 ^¶^	59	1.1 ± 0.3 ^¶^	-	235	0.08 ± 0.39	*
Vit B6 (mg)	197	1.7 ± 0.3	47	1.7 ± 0.3	-	179	1.8 ± 0.6	59	1.7 ± 1.0	-	234	0.16 ± 0.42	***
Vit C (mg)	197	76.04 ± 37.4	47	81.5 ± 48.1	-	177	61.5 ± 30.3 ^¶^	59	52.6 ± 20.0 ^¶^		233	−0.09 ± 0.66	***
Vit E (mg)	197	13.3 ± 3.8	47	13.3 ± 2.8	-	179	12.4 ± 4.7	59	12.0 ± 4.9 ^¶^	-	234	0.02 ± 0.51	-
Niacin (mg)	197	13.6 ± 3.0	47	12.7 ± 2.2	*	178	15.4 ± 5.1 ^¶^	59	14.1 ± 3.9 ^¶^	*	235	−0.02 ± 0.45	**

_a)_ 2009 data were used for 247 subjects that have follow-up data from 2012. The total follow-up number was the sum of non-OB and OB groups without missing data in 2009 and 2012, respectively. All dieary intakes were adjusted according to energy intake in 2009 and 2012, respectively. _b)_ Statistical differences between non-OB and OB groups in 2009 or 2012; *: *p* < 0.05, **; *p* < 0.01, ***; *p* < 0.001, and “-“; non-significance. _c)_ Statistical differences (paired *t*-test) between total subjects in 2009 and 2012. ^¶^; Statistical differences (*p* < 0.05) between 2009 and 2012 in non-OB or OB groups, respectively. Abbreviations: WC; waist circumferences, SBP; systolic blood pressure, DBP; diastolic blood pressure, TC; total cholesterol, TG; triacylglycerol, LDLc; LDL cholesterol, HDLc; HDL cholesterol, FBS; fasting blood glucose, CHO; carbohydrates, CHOL; dietary cholesterol, TFA; total fatty acids, SFA; saturated fatty acids, MUFA; monounsaturated fatty acids, PUFA; polyunsaturated fatty acids.

**Table 2 jpm-11-00091-t002:** Stepwise multiple regression analysis to predict the effects of the variables (the rates of change (2012–2009/2012)) on the risk of overweight/obesity according to the alleles of two *MEK6* SNPs, rs9916229 and rs756942.

SNPs	Change Rates of Variables	Major Allele			Minor Allele		
		ß	SE	*p*	ß	SE	*p*
rs 9916229 (C/A)	waist	0.191	0.028	0.000	0.108	0.038	0.006
	SBP	-	-	-	0.214	0.051	0.000
	TC	0.072	0.032	0.023			
	TG	0.030	0.008	0.000			
	FBS	−0.084	0.034	0.015			
	Protein	−0.071	0.014	0.000			
	Fat	-	-	-	0.060	0.025	0.019
	Total fatty acid	−0.138	0.039	0.001			
	MUFA	0.140	0.033	0.000			
	Fiber	-	-	-	0.105	0.032	0.002
	Veg Ca	−0.053	0.014	0.000			
	Phosphate	0.162	0.027	0.000			
	Zinc	0.133	0.023	0.000			
	*p*-value		<0.001			<0.001	
rs 756942 (C/T)	waist	0.202	0.029	0.000	0.099	0.040	0.017
	SBP	-	-	-	0.199	0.054	0.000
	TG	0.031	0.008	0.000			
	FBS	−0.083	0.034	0.016			
	Protein	−0.080	0.015	0.000			
	Fat	0.045	0.020	0.024			
	Total fatty acid	−0.119	0.040	0.003			
	MUFA	0.120	0.034	0.001			
	Calcium	-	-	-	0.067	0.029	0.025
	Veg Ca	−0.054	0.015	0.000			
	Phosphate	0.155	0.026	0.000			
	Zinc	0.110	0.022	0.000			
	Folate	-	-	-	0.041	0.017	0.016
	*p*-value		<0.001			<0.001	

## Data Availability

Data described in the manuscript will be made available upon request pending application & approval.

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
