# Peer review of "RMR-Related *MAP2K6* Gene Variation on the Risk of Overweight/Obesity in Children: A 3-Year Panel Study"

_jpm, 2021, doi:10.3390/jpm11020091_

Round 1
Reviewer 1 Report
This is an interesting study, supported by many analyses.
Main issues:
In the line 103 – 105, the author claimed that RMR was estimated using an approximating equation based on height and weight. In the line 144, the author mentioned adjusted RMR, by height and weight. These are really confusing. It seems that the RMR has been fully determined by height and weight. Same does BMI. Therefore, it seems that the relationship between BMI and RMR have been determined by body weight and height already, per the description in the section 2. Indeed, we know that RMR is affected by body compositions, what are not addressed at all in this section. Could the authors explain why they think their approach can still be valid?
Since repeated measurements (ex: RMR) were done in this cohort, how did the authors account for within person correlation analysis? If this has not been taken into account, probably it’s good to mention this as limitation in the discussion.
Minor issues:
In the study design (line 83-86), the author stated that “Using Affymetrix Genome-Wide Human SNP array 6.0 in the DNALink (Songpa-gu, Seoul, Korea), the significant genes(p<10-4) related the Low-RMR (<1055.8 kcal/day, a median) and obesity with overweight (BMI≥ 85th percentile, >19 for boys, >18.56 for girls) were detected.” Are these new findings in this study or old findings from other papers? If it’s the former, it may be better to be put in the results. If not, probably cite reference properly.
The authors need to check their grammars, spellings, and wordings. Here are two examples.
The “wild” in the sentence “A obesity rate was higher in all mutant 20 alleles of seven MEK6 SNPs than the matched wild.” may not be appropriate. Suggest using “children without XXX.”
In the introduction, the “haa” in the sentence “Based on several studies, including the Pima Indian case, that haa shown 48 that RMR has a strong genetic component, RMR might be a predictor of subsequent weight change in subjects with mutant genes that control RMR by racial or gender difference.” should be “has”?
Author Response
Main issues:
In the line 103 – 105, the author claimed that RMR was estimated using an approximating equation based on height and weight. In the line 144, the author mentioned adjusted RMR, by height and weight. These are really confusing. It seems that the RMR has been fully determined by height and weight. Same does BMI. Therefore, it seems that the relationship between BMI and RMR have been determined by body weight and height already, per the description in the section 2. Indeed, we know that RMR is affected by body compositions, what are not addressed at all in this section. Could the authors explain why they think their approach can still be valid?
ANS> I totally agree with you. Many previous studies mentioned that RMR is affected by body composition including Ht or Wt. To propose that adjusting RMR with Ht or Wt is unnecessary for children's obesity studies on line 182-183, the statistics were analyzed. Since the results are not
shown, it seems better to exclude those mention on 166 and erased them. Since the measuring of bodycomposition (i.e. fat % or lean body mass %) in children of 8~12 years old requires DEXA which was not available for economic, physical, and ethical reasons, body composition was not an option to discuss in this study. However, I mentioned in the last paragraph of “Discussion; limitation.” (405-408)
Since repeated measurements (ex: RMR) were done in this cohort, how did the authors account for within person correlation analysis? If this has not been taken into account, probably it’s good to mention this as limitation in the discussion.
ANS>. Thank you. I agree that such adjusted methods (repeated measures correlation; rmcorr) for within-individual association is needed to avoid type-I error. However, since there were only two measurements at 2009 and 2012, I did not consider type-1 error to be big enough to distort the conclusion. I did put this note into “limitation”. (405-409)
Minor issues:
In the study design (line 83-86), the author stated that “Using Affymetrix Genome-Wide Human SNP array 6.0 in the DNALink (Songpa-gu, Seoul, Korea), the significant genes(p<10-4) related the Low-RMR (<1055.8 kcal/day, a median) and obesity with overweight (BMI≥ 85th percentile, >19 for boys, >18.56 for girls) were detected.” Are these new findings in this study or old findings from other papers? If it’s the former, it may be better to be put in the results. If not, probably cite reference properly.
ANS> These findings were from our pilot study which was never published before. Since those criteria were made by grouping as the case and control in GWAS using both RMR and BMI, I made a new section 2.2 to explain this findings in “Methods” as you mentioned. (line 106-108).
The authors need to check their grammars, spellings, and wordings. Here are two examples.
- The “wild” in the sentence “A obesity rate was higher in all mutant 20 alleles of seven MEK6 SNPs than the matched wild.” may not be appropriate. Suggest using “children without XXX.”
- In the introduction, the “haa” in the sentence “Based on several studies, including the Pima Indian case, that haa shown 48 that RMR has a strong genetic component, RMR might be a predictor of subsequent weight change in subjects with mutant genes that control RMR by racial or gender difference.” should be “has”?
ANS> I corrected

Reviewer 2 Report
Thank you for the possibility to review this manuscript. Please find below some comments to the manuscript:
- Abbreviations should be defined the first time they appear in the abstract.
- Please briefly describe in the abstract the genotyping methods used in the study.
- What was the inclusion and exclusion criteria?
- Please change in the manuscript " risk of obesity" to "risk of overweight or obesity" or change the cut-off value to diagnosis of obesity from BMI ≥ 85th percentile to BMI ≥ 95th percentile.
- Additional analysis assessing separately the risk of overweight and obesity should be also performed.
- Please describe the method of DNA isolation.
- Please provide in the Method section "rs" of each analysed SNPs.
- RMR should be assessed by indirect calorimetry. The Harris & Benedict formula is not the best to calculate the RMR in children (https://pubmed.ncbi.nlm.nih.gov/8450379/). Therefore, I suggest calculating RMR base on the FAO/WHO/UNU or Schofield equation when the measurement of RMR by indirect calorimetry is not possible.
- Did the authors measure the physical activity of the study population?
- I suggest estimating total daily energy expenditure and investigate how TEE linked with analysed SNPs affected the prevalence of obesity.
- Did the authors calculate the minimum sample size or power of the study?
- Please provide the socio-economic characteristics of the study population.
- Please provide which allele you recognised as a minor/major or mutant/wild for all analysed SNPs.
- Lack of limitations and strength of the study.
- The conclusions are not clear and require redrafting.
- Please correct common typos (e.g., "haa shown", "chol-esterol").
Author Response
Abbreviations should be defined the first time they appear in the abstract.
Ans> I corrected at line 15, 21-25.
Please briefly describe in the abstract the genotyping methods used in the study.
Ans> Because of the words limitation in abstract, I mentioned it shortly. (line 19-20)
What was the inclusion and exclusion criteria?
Ans> This study has been done during the Annual Health Check period at 2009 and 2012, with the aggreements of children’s parents. We excluded the children whose parents did not agree with our
F/U Study from 2009 to 2012. Otherwise, children with consent form were all included.(line 83-85). Since “ figure-1” contains such phrases that might confuse the readers, I have edited it.
Please change in the manuscript " risk of obesity" to "risk of overweight or obesity" or change the cut-off value to diagnosis of obesity from BMI ≥ 85th percentile to BMI ≥ 95th percentile.
Additional analysis assessing separately the risk of overweight and obesity should be also performed.
ANs> I corrected “the risk of obesity” to the risk of obesity+overweight” in the text. Your second request, however, seems unavailable. Since the clinical definition of obesity in children is BMI of 95th
percentile and over, the number of children clinically defined as "obesity” would not large enough
for any parametric statistical analyses, especially considering the total number of children included
in this cohort study is less than 250.
Please describe the method of DNA isolation.
Please provide in the Method section "rs" of each analysed SNPs.
Ans> I made new section 2.2 and this note was inserted at line 98-117.
Please provide which allele you recognised as a minor/major or mutant/wild for all analysed SNPs.
Ans> The request was provided in the Text (112-113, 218-2120) and Figure 3-A with legend.
RMR should be assessed by indirect calorimetry. The Harris & Benedict formula is not the best to calculate the RMR in children (https://pubmed.ncbi.nlm.nih.gov/8450379/). Therefore, I suggest calculating RMR base on the FAO/WHO/UNU or Schofield equation when the measurement of RMR by indirect calorimetry is not possible.
Ans> I agreed with you. Since Harris & Benedict formula was used in GWAS to find gene on 2009, H & B was used in F/U study. For Korean children, many researchers have been discussed to find the best formula of RMR, but they all suggested different equations. Without the best predictive RMR to be used for Asian children, Kim et al reported HR-RMR was closed to the Korean dietary intakes (KDRI) formula followed IOM equation for elementary students. HR-RMR (1240.9± 147.4 Kcal/day) in 102 local children (9-13 years) were very similar with ours (8-11 years, 1138.8±123.4 Kcal/day). Because of Korean writing article, I did not refer before. I discussed with references of Korean and China. (Line 338-344)
Did the authors measure the physical activity of the study population?
Please provide the socio-economic characteristics of the study population.
Ans> The general information including dietary habits and degree of physical activity, was published in PLosOne (2015, DOI:10.1371/journal.pone.0120111) & International J of Obesity (2017;41:542-550). We found that the obese children seemed to have inherited obesity as judged by the BMI of their parents, but it was not correlated with parents’ education and income, health supplement intake, time of TV watching, and snack intake. Furthermore, an exercise was shown to be uncorrelated with obesity in the children because they including obese children generally took an exercise for one to three hours every day. Therefore, we did not consider those factors in F/U step.
I suggest estimating total daily energy expenditure and investigate how TEE linked with analysed SNPs affected the prevalence of obesity.
ANS> Thank you for suggesting, but since GWAS selected “MEK6” as the specific gene related to RMR and BMI, we do not know what specific genes come up when TEE gene related to BMI was analyzed.
Did the authors calculate the minimum sample size or power of the study?
Ans> Since this is the panel study that children were recruited at 2009 and the sample size was determined by follow-up same students at 2012. However, through the Levene’s normal distribution test, 247 F/W same children in both 2009 and 2012 (F/U rate of 78%) was enough to analyze minor alleles' effects on the changes of variables.
Lack of limitations and strength of the study.
Ans> This note was inserted in the text at line 405-413.
The conclusions are not clear and require redrafting.
Ans> I corrected at line 401-404.
Please correct common typos (e.g., "haa shown", "chol-esterol").
Ans> I corrected

Round 2
Reviewer 1 Report
Most points are well-addressed, and details are clear.
One minor comment:
Can you provide the values (relative frequencies) in Figure 3 (B) somewhere (ex:supplement) because it's hard to read the number based on this figure?
Author Response
Can you provide the values (relative frequencies) in Figure 3 (B) somewhere (ex:supplement) because it's hard to read the number based on this figure?
Ans> I made supplementary table 2.
Reviewer 2 Report
Please provide primer sequences.
Please provide in the supplementary materials De Finetti diagrams with Hardy-Weinberg parabola for all analysed SNPs.
Are all allele distributions consistent with the frequencies published e.g., in the 1000Genomes database?
Please correct the SNPs rs number in 113 and 222 lines (rs11654541 instead of rs1165454?).
Why did the authors not correct the results for multiple comparisons?
Predicting RMR instead of measurements by indirect calorimetry is a significant limitation of the study. Please add this information to the study limitations.
Please add the information about the generalisability of the study findings.
Author Response
Please provide primer sequences.
ANS> I made the supplementary table 1.(Line 117-118)
Please provide in the supplementary materials De Finetti diagrams with Hardy-Weinberg parabola for all analysed SNPs. Are all allele distributions consistent with the frequencies published e.g., in the 1000 Genomes database?
ANS> We requested this analysis to DNALINK (Inc, Seoul, Korea; Dr. Shin ES; esshin@dnalink.com) which GWAS and MEK6 SNPs genotyping were performed. They analyzed newly and send De Finetti plot and minor alleles frequencies in the 1000G EAS. De Finetti plot showed the range of genotype frequencies for which Hardy-Weinberg equilibrium is satisfied (the curve within the diagram). (Supplementary Figure 1)
According to allele’s frequencies in 1000G DB, MAFs of 7 MEK6 SNPs were 28.5%, 31.9%, 31.9%, 31.9%, 31.8%, 38.8%, and 37.6% with similarity of high MAFs in rs9916229 and rs756942 in this study. (Line 258-263)
Please correct the SNPs rs number in 113 and 222 lines (rs11654541 instead of rs1165454?).
ANS> I corrected this TYPO.
Why did the authors not correct the results for multiple comparisons?
ANS> I do not understand what you ask. Would you mention in detail?
Predicting RMR instead of measurements by indirect calorimetry is a significant limitation of the study. Please add this information to the study limitations.
ANS> I put this request in limitations. (453-455)
Please add the information about the generalisability of the study findings.
ANS> Without the repetition in another children population, the generalizability from this first report could not be considered. (Line 459-461)
This manuscript is a resubmission of an earlier submission. The following is a list of the peer review reports and author responses from that submission.
Round 1
Reviewer 1 Report
The manuscript by Lee et al. examined the potential differences in resting metabolic rate (RMR) in Korean children between mutant and wildtype genotypes of seven MEK6 SNPs. The investigators also used a 3-year follow-up design to explore the changes in RMR, anthropometrics, biochemical markers, and self-reported food intake. The total number of children who completed the study was 247 (unknown number of boys vs. girls). In a subgroup (n=107, including 58 boys and 59 girls), RMR and most physiological markers were measured at baseline (8 yrs) and follow-up (11 yrs), which was the major advantage of this study. The main results showed that, according to the authors’ interpretations, while “higher obesity prevalence was associated with mutant alleles of seven MEK6 SNPs”, RMRs in the mutant alleles were higher than in wild types (by about 26 kcal/day, or about 2.5%). The authors also explore many other associations. They concluded that these two SNPs might be important for energy expenditure control and could be targeted to prevent obesity. While the data collected in this cohort of children are valuable, there are, however, several key areas that can and should be improved/clarified to aid in making correct interpretations.
- The study did not have a proper power analysis to justify the number of subject needed and no significance adjustments for multiple comparisons.
- The analyses approach by the authors were not optimal to answer the question if RMRs were different between genotypes, and such differences at baseline ultimately impacted changes in body weight in 3 years. As the discussions pointed out that RMR is typically highly associated with fat-free and fat masses, or if these were not available, with either total body mass or BMI. It would make more sense to first take the baseline data (8 yrs) of whole group, find the best predictive equation for RMR from body mass, height, and sex (using multiple regression, or predictive equation), then find the difference between measured RMR to the predictive RMR for each individual. Then compare and report the differences for the mutants vs. wild types for each of the seven SNPs. Next, associate the individual differences in measured vs. predicted RMR at baseline to the changes in body weight, BMI, WC, and other biochemical markers of obesity, then see if adding the SNPs improved the association. One could also calculate measured RMR vs. predicted RMR at follow-up (11 yrs) the same way and see if the difference between mutants vs. wild types changed over time. A repeated-measure ANOVA (or ANCOVA) could also be used to determine if/which SNP(s) impacted the changes in weight changes in 3 years, and RMR was associated with such change.
- Section 2.4 RMR measurement: under what condition were these made (fasted, rested over how long? Supine? Time of the day, season? Any medication? For how long? Was all the data used? Mouthpiece or mask? What was the validity of the MedGem device in this population (PMID: 27653085 seems to show general underestimation)? Please delete all mentioning of BMR because it is more rigorous than RMR and it is unclear the oxygen consumption estimation itself would be sufficient.
- Section 2.6. Dietary habits and degree of physical activity could be very important for weight regulation, but the data were not shown anywhere.
- Lines 162-163: while adjust RMR by height, or height and weight and then associate to BMI when BMI is calculated from (thus dependent on) height and weight?
- The study did not appear to have assessed pubertal development (Tanner stage) which could be an important for the timing of weight gain.
- Figure 2: why only plot the follow-up data? Were these predicted by Harris-Benedict equation? Why not plot the measured RMR?
- Figure 3B is the only place where the obesity prevalence between genotypes and sex was shown, but the 3D plot does not allow the reader to see easily. Suggest use a table to show average and SD of BMI in each of these categories.
- Line 232-234: The last sentence (gender differences) needs to be shown by data.
- Figure 4B: Were the M to W differences in the two SNP’s due to differences in body weight (absolute RMRs are higher in heavier subjects)?
- Lines 295-303, RMR/kg body mass or fat-free mass is not recommended as a well-recognized non-zero intercept error (PMID: 22863371).
Author Response
Reviewer 1
Comments and Suggestions for Authors
The manuscript by Lee et al. examined the potential differences in resting metabolic rate (RMR) in Korean children between mutant and wildtype genotypes of seven MEK6 SNPs. The investigators also used a 3-year follow-up design to explore the changes in RMR, anthropometrics, biochemical markers, and self-reported food intake. The total number of children who completed the study was 247 (unknown number of boys vs. girls). In a subgroup (n=107, including 58 boys and 59 girls), RMR and most physiological markers were measured at baseline (8 yrs) and follow-up (11 yrs), which was the major advantage of this study. The main results showed that, according to the authors’ interpretations, while “higher obesity prevalence was associated with mutant alleles of seven MEK6 SNPs”, RMRs in the mutant alleles were higher than in wild types (by about 26 kcal/day, or about 2.5%). The authors also explore many other associations. They concluded that these two SNPs might be important for energy expenditure control and could be targeted to prevent obesity. While the data collected in this cohort of children are valuable, there are, however, several key areas that can and should be improved/clarified to aid in making correct interpretations.
- The study did not have a proper power analysis to justify the number of subject needed and no significance adjustments for multiple comparisons.
Ans> Since this is the panel study that children were recruited at the regular medical period, 2009 and the sample size was determined by follow-up same students at 2012, we can control the sample size. However, through the Levene’s normal distribution test, 247 F/W same children in both 2009 and 2012 (F/W rate of 78%) was enough to analyze minor alleles' effects on the changes of variables. (Statistics Line 161-164)
- The analyses approach by the authors were not optimal to answer the question if RMRs were different between genotypes, and such differences at baseline ultimately impacted changes in body weight in 3 years. As the discussions pointed out that RMR is typically highly associated with fat-free and fat masses, or if these were not available, with either total body mass or BMI. It would make more sense to first take the baseline data (8 yrs) of whole group, find the best predictive equation for RMR from body mass, height, and sex (using multiple regression, or predictive equation), then find the difference between measured RMR to the predictive RMR for each individual. Then compare and report the differences for the mutants vs. wild types for each of the seven SNPs. Next, associate the individual differences in measured vs. predicted RMR at baseline to the changes in body weight, BMI, WC, and other biochemical markers of obesity, then see if adding the SNPs improved the association. One could also calculate measured RMR vs. predicted RMR at follow-up (11 yrs) the same way and see if the difference between mutants vs. wild types changed over time. A repeated-measure ANOVA (or ANCOVA) could also be used to determine if/which SNP(s) impacted the changes in weight changes in 3 years, and RMR was associated with such change.
Ans> Before answering your question, I have never put “RMR” as ‘outcome’ in any of the regression analyses of this study, which thereby I have never acquired “PREDICTED RMR” from any of the regression analyses. Furthermore, “finding the best predictive equation” has never been one of this study's goals.
FIRST of all, our work aims to find the genetic environments related to RMR on obesity prevalence in children.
(1) In the pilot GWAS study, we found two genes, DNAJC6 (Ch#1) and MAP2K6 (MEK6, Ch#17), related to RMR in the children with high BMI.
(2) We focused on the effect of the MEK6 gene on pediatric obesity because there was no study of the association between MEK6 and RMR in children.
(3) We found the two minor SNPs, rs996229 and rs756942, of 7 MEK6 SNPs were strongly associated with pediatric obesity.
(4) We tested which environmental factors were related to two minor SNPs in children obesity using multiple regression and OR analyses. (Adjustment analysis was involved)
Secondly, the genetic researchers must check the difference between target gene SNPs and target variables, such as RMR or BMI. Since GWAS selected MEK6 as the specific gene related to RMR and BMI, we need not mention this assumption. However, we reported SNPs and obesity prevalence instead of BMI in children (Fig 3B), SNPs, and RMR (Fig 4B).
Thirdly, we have compared two different predictive equations: Harris-Benedict (R2=0.6079) & Mifflin St equation (10Wt+6.25Ht-5A+5 for boys, 10Wt+6.25Ht-5A-161 for girls, R2=0.5676) in 107 subjects who had a measured RMR to find the better fit. Since the Harris-Benedict equation was a better fit than the Mifflin St equation, it was used to calculate RMR (655.1+9.6Wt+1.9Ht-4.7A for girls) 66.5+13.8Wt+5Ht-6.8A for boys, as A; age) for F/W subjects. [See the Extra-Table as follows] (Line 112-127). Since measured RMR data was primarily for GWAS study in 107 subjects, we did not show this 107 RMR result in the text.
Fourthly, we did not have the approval of IRB to measure body composition like FFM or FM. (Discussion Line 366-376)
<Extra table> Comparison between the measured RMR and equations such as H-B or M-S.
|
Subject |
regression |
H-B Eq |
M-S Eq |
|
Total |
R2 |
0.6079 |
0.5676 |
|
Intercept |
222.196 |
154.3194 |
|
|
β1 |
0.6581 (P<0.001) |
0.6613(P<0.001) |
|
|
Boys |
R2 |
0.608 |
0.63 |
|
Intercept |
239.6969 (P<0.001) |
8.3300 (P=0.9861) |
|
|
β1 |
0.6437 (P<0.001) |
0.8210 |
|
|
Girls |
R2 |
0.657 |
0.7311 |
|
Intercept |
9.0146 (P=0.6816) |
222.0938 (P<0.001) |
|
|
β1 |
0.8251 (P<0.001) |
0.7293 (P<0.001) |
- Section 2.4 RMR measurement: under what condition were these made (fasted, rested over how long? Supine? Time of the day, season? Any medication? For how long? Was all the data used? Mouthpiece or mask? What was the validity of the MedGem device in this population (PMID: 27653085 seems to show general underestimation)? Please delete all mentioning of BMR because it is more rigorous than RMR and it is unclear the oxygen consumption estimation itself would be sufficient.
Ans> A detailed method for the RMR measurement was added in Section 2.4. (Line 112-127), although RMR measuring was used ONLY in GWAS PILOT STUDY, which is why I did not mention the thorough process in this paper at the beginning. PMID 27653085 using 54 adolescents, 15 years old (39 obese and 15 non-obese), MedGem is not recommended for REE's routine use because of underestimation in obesity without the gender difference. We did not test a comparison of the Indirect Methods of RMR; therefore, we did not know whether the results could be applied from 8- to 15- year old children. However, we discussed this in the limitation part with Ref 39. (Line 371-380) BMR was deleted.
- Section 2.6. Dietary habits and degree of physical activity could be very important for weight regulation, but the data were not shown anywhere.
Ans> The general information includings dietary habits and degree of physical activity, was published in PLosOne (2015, DOI:10.1371/journal.pone.0120111) & International J of Obesity (2017;41:542-550). We found that the obese children seemed to have inherited obesity as judged by the BMI of their parents, but it was not correlated with parents’ education and income, health supplement intake, time of TV watching, and snack intake. Furthermore, an exercise was shown to be uncorrelated with obesity in the children because they including obese children generally took an exercise for one to three hours every day. Therefore, we did not consider those factors in F/W step.
- Lines 162-163: while adjust RMR by height, or height and weight and then associate to BMI when BMI is calculated from (thus dependent on) height and weight?
Ans> I only left BMI-RMR correlation and deleted other graphs to help better understanding of the figure and the aim of the study.
- The study did not appear to have assessed pubertal development (Tanner stage) which could be an important for the timing of weight gain.
ANS> We could not get consent form from the parents when we included Tanner stage evaluation in the study. Therefore, we recruited boys and girls at 8, who neither shows 2nd sexual characteristics nor menstruation.
- Figure 2: why only plot the follow-up data? Were these predicted by Harris-Benedict equation? Why not plot the measured RMR?
ANS> All data used in this study was described as the change rate for three years [(2012-2009)/2012) in the same F/W children. Yes, we used the Harris-Benedict equation and mentioned why we used H-B Eq for 247 F/W children. The measured RMR was only used in the GWAS study to find the Genes related to RMR in 107 children at the baseline. See the answer for Q 2.
- Figure 3B is the only place where the obesity prevalence between genotypes and sex was shown, but the 3D plot does not allow the reader to see easily. Suggest use a table to show average and SD of BMI in each of these categories.
ANS> Since the study subjects were prepubertal children, the numeric value of BMI is NOT the standard of diagnosing obesity but the percentile(85th and over) at his/her age of growth curve, which means that “obesity” itself in children already involves the concept of mean/SD of BMI. Therefore, we used obesity “prevalence” instead of the numeric value of BMI in order to distinguish “obesity” from “non-obesity.” Furthermore, since the MEK6 gene related to RMR was screened in obesity compared to non-OB, BMI levels were not significantly different according to MEK6 SNPs, respectively. Therefore, we used the obesity prevalence in SNPs and found that the obesity prevalence was higher in boys having mutants of rs9916229 and rs756942 compared to girls. (gender significances) It seemed better to show the chi-sq result in Figure to understand instead of the Table as follows. Therefore, we modified this picture to be easily understood and explained in text Line 202-206.
- Line 232-234: The last sentence (gender differences) needs to be shown by data.
ANS> The gender difference was shown in data of Fig 3 (B). See ANS of #8
- Figure 4B: Were the M to W differences in the two SNP’s due to differences in body weight (absolute RMRs are higher in heavier subjects)?
ANS> We already confirmed that RMR was highly correlated to BMI (not weight) in Fig 2, but we did not show a significance of BMI in rs9916229 and rs756942. (ANS for #8.) In Fig 4B, we could not compare the RF of obesity in 3 genotypes because of small subjects in mutant homozygotes (n=9 to 13). Although RMR was higher in mutant than wild of all MEK6 SNPs, we found significant RMR differences in 3 genotypes of only two SNPs. Since we found the impact of mutant homozygotes of two MEK6 SNPs on RMR to modulate the children obesity, we look forward to environmental factors related to two SNPs, particularly. (Table 2 & Fig 5) This is our final purpose.
- Lines 295-303, RMR/kg body mass or fat-free mass is not recommended as a well-recognized non-zero intercept error (PMID: 22863371).
ANS> We discussed this suggestion in Discussion with Ref 26: Line 324-238.

Reviewer 2 Report
The manuscript by Lee et al. describes based on their pilot study association of the MAP2K6 gene and resting metabolic rate (RMR) among Korean school children aged 8-10 years. The three-year follow-up study showed that obesity increased from 19.4% to 25.5%, and change rates of different variables were recorded. RMRs increased in mutant alleles of 2SNPs and control variables prevents obesity and have two mutants of MEK6 as reference validating study. There are a few concerns as follows.
- The changes in the variable of obesity were recorded in Table 1. The total number of participants differs between each variable. Suppose the blood sample is collected from all the participants why there is a difference between variable participants number (N). For example, TC, TG in the 2009 Non-OB group N=199 and FBS variable with N=194 if the blood sample collected from N=199 participants why the variable FBS contains 194 participants. This needs to be explained in the methodology of the discrepancy in the participant number for each variable. This may lead to variation in the results.
- Also, the total number of participants CHO, protein variable participants in Non-OB (N=179), OB+overweight (N=59) group combined participants is not matching with Change rate ‘N’ numbers (N=235). The total number differs, which may have an impact on the results presented in the table.
- Figure 5 provides the significant (*, **) against which group either low or mid group comparison.
- The discussion needs to be modified to reflect the difference of variables in table 1.
Author Response
Reviewer 2
Comments and Suggestions for Authors
The manuscript by Lee et al. describes based on their pilot study association of the MAP2K6 gene and resting metabolic rate (RMR) among Korean school children aged 8-10 years. The three-year follow-up study showed that obesity increased from 19.4% to 25.5%, and change rates of different variables were recorded. RMRs increased in mutant alleles of 2SNPs and control variables prevents obesity and have two mutants of MEK6 as reference validating study. There are a few concerns as follows.
- The changes in the variable of obesity were recorded in Table 1. The total number of participants differs between each variable. Suppose the blood sample is collected from all the participants why there is a difference between variable participants number (N). For example, TC, TG in the 2009 Non-OB group N=199 and FBS variable with N=194 if the blood sample collected from N=199 participants why the variable FBS contains 194 participants. This needs to be explained in the methodology of the discrepancy in the participant number for each variable. This may lead to variation in the results.
ANS> The F/W number would be the sum of non-OB and OB in 2012 unless there were no missing data. Since there were some missing data in 2009 and 2012, respectively, the F/W number may be different from the sum of non-OB and OB in 2012. We explained this in the legend of Table 1. N=162 was a typo, and the correct number is n=208 for FBS. Thank you for finding the error. Otherwise, FBS in blood, was not the risk levels, was not different between non-OB and OB.
- Also, the total number of participants CHO, protein variable participants in Non-OB (N=179), OB+overweight (N=59) group combined participants is not matching with Change rate ‘N’ numbers (N=235). The total number differs, which may have an impact on the results presented in the table.
ANS> Since the change rates of variables were calculated by (2012-2009)/2012, subjects who had the data of both 2009 and 2012 must be counted. We would want the numbers were matched between 2009 and 2012 without missing data. However, there was no statistics error.
- Figure 5 provides the significant (*, **) against which group either low or mid group comparison.
ANS> We found the mistakes on Fig 5 that the statistical significance compared to Q1 (Low) should be also described in Q2 (Mild) and all was corrected. But the description in Text was right on Line 264-277.
- The discussion needs to be modified to reflect the difference of variables in table 1.
ANS> In the other children cohort set, we found the same patterns of blood profiles and dietary intakes for three years of the panel study. (PLosOne, https://doi.org/10.1371/journal.pone. 0120111) The general information, such as dietary habits and degree of physical activity, was published in PLosOne as the baseline data in a cross-sectional study. (Line 304-308) Therefore, we focused on genetic environments related to RMR on energy expenditure to modulate the obesity prevalence in this study.
We aimed to find the genetic environments related to RMR on obesity prevalence in children using the rate of the change of variables for 3 years and MEK6 SNPs’ that we found the specific gene in pilot GWAS. (Line 315-317) Although the paper reported “the insulin and HOMA-IR levels were more favorable in high RMR/Kg(≥20) than low RMR” in DM patients, it could not be compared with this children’s study. Since TC, TG, and FBS variables were not affected by a mutant of MEK6 SNPs, we did not discuss. A few data with dietary intakes related to MEK6 SNPs were discussed. (Line 352-364)
Since this study is the first report that MEK6 SNPs are related to energy expenditure on obesity in children, we expect the general information like Table 1 will be the basis of future study.

Round 2
Reviewer 1 Report
Thank you for the revision. There are still some key misunderstandings.
- Measured RMR: given the errors in MedGem (lines 377-378), it cannot be considered as a "criterion" measure of RMR (line 115). Also, it only measures VO2 (oxygen consumption), and the RMR formula on line 118 is incorrect (should be VO2 and not CO2). If the sentence on line 119-120 is correct, does that mean the "large children" were less compliant? Does this introduce a bias in the study results - less compliant lead to higher RMR, or less data used? If the MedGem's measured RMR was highly correlated to predicted RMR, could it be due to the MedGem using a weight and/or height associated model (this has been suspected previously)? What if you enter another weight and height, or even enter zeros, does the MedGem give you another RMR?
- The over-reliance on R values: lines 121-126, were either predictive equations validated in Asian adolescents? Please show the Bland-Altman plots, and include the Schofield age-specific RMR predictive equation. R values alone does not reflect how accurate predictive equations are. For this type of study, it would be critical to determine if the accuracy is comparable for lower and higher RMRs.
- Line 192: "correlation between BMI and Wt or Ht were not signficant" is probably not right - Wt is the numerator of BMI. Not clear why this paragraph was try to say - if the RMR is "adjusted" (not adjuted"), the correct procedure should be a multiple regression (stepwise or similar) with measured independent factors (wt, ht, age, etc).
- My previous suggestion of develop such a regression was not meant to publish another RMR predictive equation, but to analyze differences between individual vs. averaged RMR in this group, and compare if the SNP's have different RMR's. And since you have good follow-up data, it would be valuable to explore if the delta's between individual and group average RMR predict the weight gain (similar to the Ravussin et al. paper, Ref 8).
- Given the recognition of the RMR/kg problem (Line 329-331), why still include the calculations of RMR/BMI or /kg on lines 316-318?